# Lagrange form of the nonlinear Schrödinger equation for low-vorticity waves in deep water

Anatoly Abrashkin[1] and Efim Pelinovsky[2,3]

[1] National Research University Higher School of Economics (HSE),
25/12 Bol'shaya Pecherskaya str., Nizhny Novgorod, 603155, Russia

[2] Institute of Applied Physics RAS, 46 Ulyanov str., Nizhny Novgorod, 603950, Russia

[3] Nizhny Novgorod State Technical University n.a. R. Alekseev, 24 Minin str., Nizhny Novgorod, 603950, Russia

The nonlinear Schrödinger (NLS) equation describing the propagation of weakly rotational wave packets in an infinitely deep fluid in Lagrangian coordinates has been derived. The vorticity is assumed to be an arbitrary function of Lagrangian coordinates and quadratic in the small parameter proportional to the wave steepness. The vorticity effects manifest themselves in a shift of the wavenumber in the carrier wave as well as in variation of the coefficient multiplying the nonlinear term. In the case of the dependence of vorticity on the vertical Lagrangian coordinate only (the Gouyon waves), the shift of the wavenumber and the respective coefficient are constant. When the vorticity is dependent on both Lagrangian coordinates, the shift of the wavenumber is horizontally inhomogeneous. There are special cases (e.g., Gerstner waves) when the vorticity is proportional to squared wave amplitude and non-linearity disappears, thus making the equations for wave packet dynamics linear. It is shown that the NLS solution for weakly rotational waves in the Eulerian variables may be obtained from the Lagrangian solution by simply changing the horizontal coordinates.

Key words: nonlinear Schrödinger equation, vorticity, water waves

## 1 Introduction

The nonlinear Schrödinger (NLS) equation was first derived by Zakharov in 1967 (English edition, Zakharov, 1968) who used the Hamiltonian formalism for description of wave propagation in deep water; see also Benney and Newell (1967). Hashimoto and Ono (1972) and Davey (1972) obtained the same result independently. Like Benney and Newell (1967) they used the method of multiple scale expansions in Euler coordinates. Yuen and Lake (1975), in turn, derived the NLS equation on the basis of the averaged Lagrangian method. Benney and Roskes (1969) extended those two-dimensional theories to the case of three-dimensional wave perturbations in a finite depth fluid and obtained equations that are now known as the Davey-Stewartson equations. In this particular case the equation proves the existence of transverse instability of a plane wave which is much stronger than a longitudinal one. This circumstance diminishes the role and

meaning of the NLS equation for sea applications. Meanwhile, the 1-D NLS
equation has been successfully tested many times in laboratory wave tanks and
natural observations were compared with numerical calculations in the framework
of this equation.
In all the cited papers wave motion was considered to be potential. However,
wave formation and propagation frequently occur against the background of a
shear flow possessing vorticity. Wave train modulations upon arbitrary vertically
sheared currents were studied by Benney and Maslowe (1975). Using the method
of multiple scales, Johnson (1976) examined slow modulation of a harmonic wave
moving at the surface of an arbitrary shear flow with velocity profile $U(y)$, where
$y$ is vertical coordinate. He derived the NLS equation with coefficients which
depend in a complicated way on a shear flow (Johnson, 1976). Oikawa et al.
(1985) considered the properties of instability of weakly nonlinear three-
dimensional wave packets in the presence of a shear flow. Their simultaneous
equations reduce to the known NLS equation for the case of purely two-
dimensional wave evolution. Li et al. (1987) and Baumstein (1998) studied the
modulation instability of the Stokes wave-train and derived an NLS equation for an
uniform shear flow in deep water, when $U(y) = \Omega_0 y$ and $\Omega_z = \Omega_0$ is constant
vorticity ($z$ is the horizontal coordinate normal to the $x, y$ plane of the flow; the
wave propagates in the $x$ direction).
Thomas et al. (2012) generalized their results for a finite-depth fluid and
confirmed that a linear shear flow may significantly modify the stability properties
of weakly nonlinear Stokes waves. In particular, for the waves propagating in the
direction of the flow, the Benjamin-Feir (modulational) instability can vanish in
the presence of positive vorticity ($\Omega_0 < 0$) for any depth.
In the traditional Eulerian approach to the propagation of weakly nonlinear
waves against the background current, a shear flow determines vorticity in a zero
approximation. Depending on the flow profile $U(y)$ it may be arbitrary and equal
to $-U'(y)$. At the same time, the vorticity of wave perturbations $\Omega_n, n \geq 1$, i.e. the
vorticity in the first and subsequent approximations in the wave steepness
parameter $\varepsilon = kA_0$ ($k$ is wavenumber and $A_0$ is wave amplitude) depends on its
form. In Eulerian coordinates the vorticity of wave perturbations is a function not
only of $y$, but of $x$ and $t$ variables as well. Plane waves on a shear flow with a linear
vertical profile are regarded to be an exception (Li et al., 1987; Baumstein, 1998;
Thomas et al., 2012). For such waves the vorticity is constant in a zero
approximation, and all the vorticities in wave perturbations are equal to zero. For
an arbitrary vertical profile of the shear flow (Johnson, 1976), expressions for the
functions $\Omega_n$ can be hardly predicted even qualitatively.
The Lagrangian method allows applying a different approach. In the plane
flow the vorticity of fluid particles is preserved and can be expressed via
Lagrangian coordinates only. Thus, not only the vertical profile of the shear flow
defining the vorticity in a zero approximation, but the expressions for the vorticity
of the following orders of smallness can also be arbitrary. The expression for the
vorticity is written in the form
$$\Omega(a,b) = -U'(b) + \sum_{n \geq 1} \varepsilon^n \Omega_n(a,b),$$

where $a, b$ are the horizontal and vertical Lagrangian coordinates, respectively,
$U(b)$ is the vertical profile of the shear flow, and particular conditions for defining
the $\Omega_n$ functions can be found. For the given shear flow this approach allows
studying wave perturbations under the most general law of distribution of
vorticities $\Omega_n$. In the present paper we do not consider shear flow and vorticity in
the linear approximation $(U = 0; \Omega_1 = 0)$, whereas vorticity in the quadratic
approximation is an arbitrary function. This corresponds to the rotational flow
proportional to $\varepsilon^2$. We can define both the shear flow and the localized vortex.
The dynamics of plane wave trains on the background flows with arbitrary
low vorticity has not been studied before. The idea to study wave trains with
quadratic (with respect to the wave steepness parameter) vorticity was realized
earlier for the spatial problems in the Euler variables. Hjelmervik and Trulsen
(2009) derived the NLS equation for vorticity distribution
$$\Omega_y / \omega = O(\varepsilon^2); \quad (\Omega_x, \Omega_z) / \omega = O(\varepsilon^3),$$

where $\omega$ is wave frequency. The vertical vorticity of wave perturbations exceeds
the other two vorticity components by a factor of ten. This vorticity distribution
corresponds to the low (of order $\varepsilon$) velocity of the horizontally inhomogeneous
shear flow. Hjelmervik and Trulsen (2009) used the NLS equation to study the
statistics of rogue waves on narrow current jets, and Onorato et al. (2011) used that
equation to study the opposite flow rogue waves. The effect of low vorticity ($\varepsilon^2$
order of magnitude) in the paper by Hjelmervik and Trulsen (2009) is reflected in
the NLS equation. This fact, like the NLS nonlinear term for plane potential waves,
may be attributed to the presence of an average current non-uniform over the fluid
depth.
Colin et al. (1995) considered the evolution of three-dimensional vortex
disturbances in a finite-depth fluid for a different type of vorticity distribution:

$$\Omega_y = 0; \quad (\Omega_x, \Omega_z) / \omega = O(\varepsilon^2)$$

and reduced the problem to a solution of the Davey-Stewartson equations by
means of the multiple scale expansion method in Eulerian variables. In this case,
vorticity components are calculated after the solution of the problem. Similarly to
the traditional Eulerian approach (Johnson, 1976) the form of the quadratic
vorticity distribution is very special and does not cover all of its numerous possible
distributions.
In this paper we consider the plane problem of nonlinear wave packet
propagating in an ideal incompressible fluid with the following form of vorticity
distribution

$$\Omega_z / \omega = O(\varepsilon^2).$$

In contrast to Hjelmervik and Trulsen (2009), Onorato et al. (2011) and Colin et al.
(1996), the flow is two-dimensional ($\Omega_x = \Omega_y = 0$). The propagation of a packet of
potential waves gives rise to a weak counterflow underneath the free water surface
with velocity proportional to the square of the wave steepness (McIntyre, 1982). In
the considered problem this potential flow is superimposed with the rotational one
of the same order of magnitude. This results in appearance of an additional term in
the NLS equation and in change of the coefficient in the nonlinear term. So, the
difference from the NLS solutions derived for a strictly potential fluid motion was
revealed.
The examination is made in the Lagrangian variables. The Lagrangian
variables are rarely used in fluid mechanics because of a more complex type of
nonlinear equations in Lagrangian form. However, when considering the vortex-
induced oscillations of a free fluid surface, the Lagrangian approach has two major
advantages. First, unlike the Euler description method the shape of the free surface
is known and is determined by the condition of the equality to zero ($b = 0$) of the
vertical Lagrangian coordinate. Second, the vortical motion of liquid particles is
confined within the plane and is a function of Lagrangian variables $\Omega_z = \Omega_z(a,b)$,
so the type of vorticity distribution in the fluid can be preset. The Eulerian
approach does not allow this. In this case, the second-order vorticity is defined as a
known function of Lagrangian variables.
Here, hydrodynamic equations are solved in Lagrangian form by the
multiple scale expansion method. A nonlinear Schrödinger equation with variable
coefficients is derived. Possible ways of reducing it to the NLS equation with
constant coefficients are studied.
The paper is organized as follows. Section 2 describes the Lagrangian
approach to studying wave oscillations at the free surface of a fluid. The zero of
the Lagrangian vertical coordinate corresponds to the free surface, thus simplifying
formulation of the pressure boundary conditions. The specific feature of the
proposed approach is introduction of a complex coordinate of a fluid particle
trajectory. In Section 3 a nonlinear evolution equation is derived on the basis of the
method of multiple scale expansion. Different solutions of the NLS equation
adequately describing various examples of vortex waves are considered in Section
4. The transform from the Lagrangian coordinates to the Euler description of
solutions of the NLS equation is shown in Section 5. Section 6 summarizes the
obtained results.

**2 Basic equations in Lagrangian coordinates**

Consider the propagation of a packet of gravity surface waves in a rotational
infinitely deep fluid. 2D hydrodynamic equations of an incompressible inviscid
fluid in Lagrangian coordinates have the following form (Lamb, 1932; Abrashkin
and Yakubovich, 2006; Bennett, 2006):
$$\frac{D(X,Y)}{D(a,b)} = [X,Y] = 1, \tag{1}$$

$$X_{tt}X_a + \left(Y_{tt} + g\right)Y_a = -\frac{1}{\rho}p_a, \tag{2}$$

$$X_{tt}X_b + \left(Y_{tt} + g\right)Y_b = -\frac{1}{\rho}p_b, \tag{3}$$


where $X,Y$ are the horizontal and vertical Cartesian coordinates and $a,b$ are the
horizontal and vertical Lagrangian coordinates of fluid particles, $t$ is time, $\rho$ is
fluid density, $p$ is pressure, $g$ is acceleration due to gravity, the subscripts mean
differentiation with respect to the corresponding variable. The square brackets
denote the Jacobian. The $b$ axis is directed upwards, and $b = 0$ corresponds to the
free surface. Equation (1) is a volume conservation equation. Equations (2) and (3)
are momentum equations. The geometry of the problem is presented in Fig. 1.

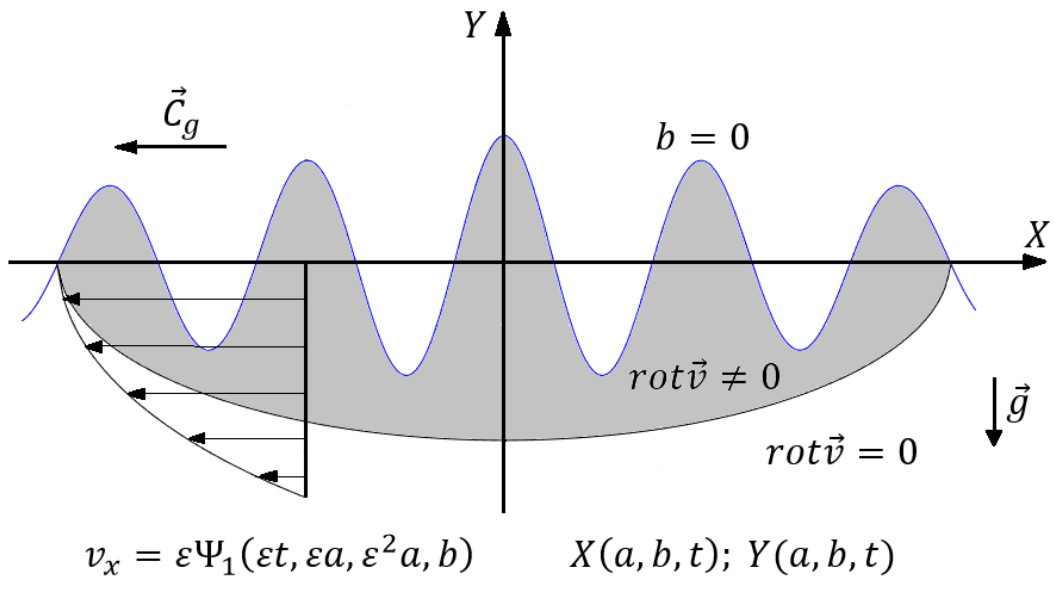

**Fig. 1. Problem geometry: $v_x$ is average current.**
194       Making use of cross differentiation it is possible to exclude pressure and to
obtain the condition of conservation of vorticity along the trajectory (Lamb, 1932;
Abrashkin and Yakubovich, 2006; Bennett, 2006):

$$X_{ta}X_b + Y_{ta}Y_b - X_{tb}X_a - Y_{tb}Y_a = \Omega(a,b). \tag{4}$$


This equation is equivalent to the momentum equations (2) and (3) but involves explicit vorticity of liquid particles, $\Omega$, which in case of two-dimensional flows is the function of Lagrangian coordinates only.

We introduce a complex coordinate of a fluid particle trajectory $W = X + iY$ $(\overline{W} = X - iY)$, the overline means complex conjugation. In the new variables Eqs. (1) and (4) take on the form

$$[W, \overline{W}] = -2i, \tag{5}$$

$$\text{Re}[W_t, \overline{W}] = \Omega(a,b), \tag{6}$$

After simple algebraic manipulations Eqs. (2) and (3) reduce to the following single equation

$$W_{tt} = -ig + i\rho^{-1}[p, W]. \tag{7}$$

Equations (5) and (6) will be further used to find the coordinates of complex trajectories of fluid particles, and Eq. (7) determines the pressure of the fluid. The boundary conditions are the non-flowing condition at the bottom ( $Y_t \to 0$ at $b \to -\infty$ ) and constant pressure at the free surface (at $b = 0$).

The Lagrangian coordinates mark the position of fluid particles. In the Eulerian description the displacement of the free surface $Y_s(X,t)$ is calculated in an explicit form, but in the Lagrangian description it is defined parametrically by the following equalities: $Y_s(a,t) = Y(a,b=0,t)$; $X_s(a,t) = X(a,b=0,t)$, where the Lagrangian horizontal coordinate $a$ plays the role of parameter. Its value along the free surface $b = 0$ varies in the $(-\infty;\infty)$ range. In Lagrangian coordinates the function $Y_s(a,t)$ defines the displacement of the free surface.

**3 Derivation of evolution equation**

Let us represent the function $W$ using the multiple scales method in the following form

$$W = a_0 + ib + w(a_l, b, t_l), \quad a_l = \varepsilon^l a, \quad t_l = \varepsilon^l t; \quad l = 0,1,2, \tag{8}$$

where $\varepsilon$ is the small parameter of wave steepness. All unknown functions and the given vorticity can be represented as a series in this parameter:

$$w = \sum_{n=1} \varepsilon^n w_n; \quad p = p_0 - \rho g b + \sum_{n=1} \varepsilon^n p_n; \quad \Omega = \sum_{n=1} \varepsilon^n \Omega_n(a,b). \tag{9}$$

In the formula for the pressure, the term with hydrostatic pressure is selected, $p_0$
is the constant atmospheric pressure at the fluid surface. The representations (8)
and (9) are substituted into Eqs. (5)-(7).

**3.1 Linear approximation**

In a first approximation in the small parameter we have the following system of
equations

$$\mathrm{Im}\left(iw_{1a_0} + w_{1b}\right) = 0, \tag{10}$$

$$\mathrm{Re}\left(iw_{1a_0} + w_{1b}\right)_{t_0} = -\Omega_1, \tag{11}$$

$$w_{1t_0t_0} + \rho^{-1}\left(p_{1a_0} + ip_{1b}\right) = igw_{1a_0}. \tag{12}$$


The solution satisfying the continuity equation (10) and the equation of
conservation of vorticity (11) describes a monochromatic wave (for definiteness,
we consider the wave propagating to the left) and the average horizontal current

$$w_1 = A(a_1, a_2, t_1, t_2)\exp\left[i(ka_0 + \omega t_0) + kb\right] + \psi_1(a_1, a_2, b, t_1, t_2); \quad \Omega_1 = 0. \tag{13}$$


Here $A$ is the complex amplitude of the wave, $\omega$ is its frequency, and $k$ is the
wave number. The function $\psi_1$ is real and will be found in the next
approximation.
261   The substitution of solution (13) into Eq. (12) yields the equation for the
pressure

$$\rho^{-1}\left(p_{1a_0} + ip_{1b}\right) = \left(\omega^2 - gk\right)A\exp\left[i(ka_0 + \omega t_0) + kb\right], \tag{14}$$


which is solved analytically

$$p_1 = -\mathrm{Re}\,\frac{i\left(\omega^2 - gk\right)}{k}\rho A\exp\left[i(ka_0 + \omega t_0) + kb\right] + C_1\left(a_1, a_2, t_1, t_2\right), \tag{15}$$


where $C_1$ is an arbitrary function. The boundary condition at the free surface is
$p_1|_{b=0} = 0$, which leads to $\omega^2 = gk$ and to $C_1 = 0$. Thus, in the first approximation
the pressure correction $p_1$ is equal to zero.

**3.2 Quadratic approximation**

The equations of the second order of the perturbation theory can be written as follows

$$\mathrm{Im}\left(iw_{2a_0} + w_{2b} + iw_{1a_1} - w_{1a1}\overline{w_{1b}}\right) = 0, \tag{16}$$

$$\mathrm{Re}\left[iw_{2t_0a_0} + w_{2t_0b} + i\left(w_{1t_0a_1} + w_{1t_1a_0}\right) - w_{1t_0a_0}\overline{w_{1b}} + w_{1t_1b} + w_{1t_0b}\overline{w_{1a_0}}\right] = -\Omega_2, \tag{17}$$

$$w_{2t_0t_0} + \rho^{-1}\left(p_{2a_0} + ip_{2b}\right) = ig\left(w_{2a_0} + w_{a_1}\right) - 2w_{1t_1t_0}. \tag{18}$$

By substituting expression (13) for $w_1$ into Eq. (16) we obtain

$$\mathrm{Im}\left[iw_{2a_0} + w_{2b} - i\left(k\psi_{1b}A - A_{a_1}\right)\exp\left[i\left(ka_0 + \omega t_0\right) + kb\right] - ik^2|A|^2 e^{2kb} + i\psi_{1a_1}\right] = 0, \tag{19}$$

which is integrated as follows

$$w_2 = i\left(kA\psi_1 - bA_{a_1}\right)\exp\left[i\left(ka_0 + \omega t_0\right) + kb\right] + \psi_2 + if_2, \tag{20}$$

where $\psi_2, f_2$ are the functions of slow coordinates and Lagrangian vertical coordinate $b$ and

$$f_{2b} = k^2|A|^2 \exp 2kb - \psi_{1a_1}, \tag{21}$$

$\psi_2$ is an arbitrary real function. It will be determined in a solution in the cubic approximation.

When (13) and (20) are substituted into (17), the sum of the terms containing the exponential factor becomes equal to zero, and the remaining terms satisfy the equation

$$\psi_{1t_1b} = -2k^2\omega|A|^2 \exp(2kb) - \Omega_2. \tag{22}$$

The expression for the function $\psi_1$ can be found by a simple integration. It should be emphasized that the vorticity in the second approximation, that is part of Eq. (22), is an arbitrary function of slow horizontal and vertical Lagrange coordinates, so that $\Omega_2 = \Omega_2(a_1, a_2, b)$.

Taking into account the solutions in the first two approximations we can write Eq. (18) as

$$\rho^{-1}\left(p_{2a_0} + ip_{2b}\right) = i\left(gA_{a_1} - 2\omega A_{t_1}\right)\exp\left[i\left(ka_0 + \omega t_0\right) + kb\right] + ig\psi_{1a_1}. \tag{23}$$

Its solution determines the pressure correction

$$p_2 = \text{Re}\left[\frac{1}{k}\left(gA_{a_1} - 2\omega A_{t_1}\right)\exp\left[i\left(ka_0 + \omega t_0\right) + kb\right]\right] + \rho g\int_0^b \psi_{1a_1} db + C_2\left(a_1, a_2, t_1, t_2\right) \quad (24)$$

The integration limits in the penultimate term are chosen so that this integral term equals zero at the free surface. Due to the boundary condition for pressure ($p_2(b=0)=0$), $C_2=0$, and

$$A_{t_1} - c_g A_{a_1} = 0; \quad c_g = \frac{g}{2\omega} = \frac{1}{2}\sqrt{\frac{g}{k}}, \quad (25)$$

where $c_g$ is the group velocity of wave propagation in deep water, which in this approximation is independent of fluid vorticity. As was expected, in this approximation the wave moves with group velocity $c_g$ to the left (the "minus" sign in Eq. (25)).

### 3.3 Cubic approximation

The equation of continuity and the condition of conservation of vorticity in the third approximation are written in the form

$$\text{Im}\left[iw_{2a_0} + w_{3b} + i\left(w_{1a_2} + w_{2a_1} + w_{2a_0}\right) - \left(w_{1a_1} + w_{2a_2}\right)\overline{w_{1b}} - w_{1a_0}\overline{w_{2b}}\right] = 0, \quad (26)$$

$$\text{Re}\left[iw_{3t_0a_0} + w_{3t_0b} + i\left(w_{1t_2a_0} + w_{1t_1a_1} + w_{1t_0a_2} + w_{2t_1a_0} + w_{2t_0a_1}\right) + w_{1t_2b} - \overline{w_{2b}}w_{1t_0a_0} - \right.$$

$$\quad (27)$$

$$\left. + w_{2t_1b} - w_{1b}\left(w_{1t_0a_1} + w_{1t_1a_0} + w_{2t_0a_0}\right) + \overline{w_{1a_0}}\left(w_{1t_1b} + w_{2t_0b}\right) + w_{1t_0b}\left(\overline{w_{1a_1}} + \overline{w_{2a_0}}\right)\right] = -\Omega_3.$$

We substitute the solutions in the first and second approximations into the simultaneous equations

$$\text{Im}\left[iw_{3a_0} + w_{3b} + i\left(\psi_{1a_2} + \psi_{2a_1}\right) + 2k(kb+1)A\overline{A_{a_1}}e^{2b} + G_b e^{i\left(ka_0 + \omega t_0\right) + kb}\right] = 0, \quad (28)$$


$$\text{Re}\left\{\left[iw_{3a_0} + w_{3b} + \left(G_b + 2k\psi_{1t_1b}\omega^{-1}A\right)e^{i\left(ka_0 + \omega t_0\right)+ kb}\right]_{t_0} + \psi_{2t_1b} + \psi_{1t_2b} + \right.$$

$$\left. + i\omega k(4kb+5)A\overline{A_{a_1}}e^{2kb}\right\} = -\Omega_3, \tag{29}$$


$$G = ibA_{a_2} + \frac{b^2}{2}A_{a_1a_1} - (kb+1)\psi_1 A_{a_1} - \left(ik\psi_2 + kf_2 - \frac{k^2}{2}\psi_1^2\right)A. \tag{30}$$


We seek a solution for the third approximation in the following form

$$w_3 = \left(G_1 - G\right)e^{i\left(ka_0+\omega t_0\right)+kb} + G_2 e^{-i\left(ka_0+\omega t_0\right)+kb} + \psi_3 + if_3, \tag{31}$$


where $G_1, G_2, \psi_3, f_3$ are functions of slow coordinates and $b$. By substituting this
expression into (28) and (29) we immediately find

$$f_{3b} + \psi_{2a_1} + \psi_{1a_2} + k(kb+1)\left(A\overline{A_{a_1}} - \overline{A}A_{a_1}\right)e^{2kb} = 0, \tag{32}$$


$$\psi_{2t_1b} + \psi_{1t_2b} + \frac{1}{2}(4kb+5)\omega k\left(A\overline{A_{a_1}} - \overline{A}A_{a_1}\right)e^{2kb} = -\Omega_3. \tag{33}$$


The function $\psi_2$ according to Eq. (33) is determined by known solutions for $A$
and $\psi_1$, and by the given distribution of $\Omega_3$. The expression for the function $f_3$
is then derived from Eq. (32). These functions determine the horizontal and
vertical average motion, respectively. But in this approximation they are not
included in the evolution equation for the wave envelope. The function $\psi_3$ will be
found in the next approximation.
When solving (28) and (29) we have found

$$G_1 = -k\omega^{-1}\psi_{1t_1}A; \quad G_2 = k\omega^{-1}\left(2ke^{-2kb}\int_{-\infty}^{b}\psi_{1t_1}e^{2kb'}db' - \psi_{1t_1}\right)\overline{A}. \tag{34}$$


These relationships should be substituted into Eq. (7), which in this approximation
has the form

$$w_{3t_0t_0} - igw_{3a_0} = i\rho^{-1}\left[i\left(p_{2a_1} + p_{3a_0}\right) - p_{3b} - p_{2b}w_{1a_0} + \rho g\left(w_{1a_2} + w_{2a_1}\right)\right] -$$

$$- 2w_{1t_2t_0} - w_{1t_1t_1} - 2w_{2t_0t_1}. \tag{35}$$


Taking into account (13), (20), (24), (31) and (34) we rewrite it as follows

$$\rho^{-1}\left(p_{3a_0} + ip_{3b}\right) = \left(-2i\omega\frac{\partial A}{\partial t_2} + ig\frac{\partial A}{\partial a_2} - \frac{\partial^2 A}{\partial t_1^2} + 2\omega k\psi_{1t_1}A\right)e^{i\left(ka_0 + \omega t_0\right)+kb} + \qquad (36)$$

$$+ 2\omega^2 G_2\overline{A}e^{-i\left(ka_0+\omega t_0\right)+kb} + ig\left(\psi_{2a_1} + \psi_{1a_2}\right) + I; \quad I = -g\left(f_{2a_1} - \int_b^0\psi_{1a_1a_1}db\right) - \psi_{t_1t_1}.$$

By virtue of the relationships (21), (22) and (25) the derivative of $I$ along the
vertical Lagrangian coordinate is zero ($I_b = 0$), so $I$ is the only function of the
slow coordinates and time - $a_l, t_l, l \geq 1$. The contribution of the term $I\left(a_l, t_l\right) \neq 0$ to
the pressure is complex, so it demands $I = 0$.
379   The solution of Eq. (36) yields the expression for the pressure perturbation
in the third approximation:

$$\frac{p_3}{\rho} = \mathrm{Re}\,ik^{-1}\left(2i\omega\frac{\partial A}{\partial t_2} - ig\frac{\partial A}{\partial a_2} + \frac{\partial^2 A}{\partial t_1^2} - 4\omega k^2 Ae^{-2kb}\int_{-\infty}^b\psi_{1t_1}e^{2kb'}\,db'\right)e^{i\left(ka_0+\omega t_0\right)+kb} +$$

$$+ \rho g\int_0^b\left(\psi_{2a_1} + \psi_{1a_2}\right)db'. \qquad (37)$$

In Eq. (37) the integration limits for the second integral term have been preset to
satisfy the boundary condition at the free surface (the pressure $p_3$ should turn to
zero). Then the factor before the exponent should be equal to zero:

$$2i\omega\frac{\partial A}{\partial t_2} - ig\frac{\partial A}{\partial a_2} + \frac{\partial^2 A}{\partial t_1^2} - 4\omega k^2 A\int_{-\infty}^0\psi_{1t_1}e^{2kb}\,db = 0. \qquad (38)$$

390   By introducing the "running" coordinate $\zeta_2 = a_2 + c_g t_2$ we can reduce Eq.
(38) to a compact form

$$i\frac{\partial A}{\partial a_2} - \frac{k}{\omega^2}\frac{\partial^2 A}{\partial t_1^2} + \frac{4k^3 A}{\omega}\int_{-\infty}^0\psi_{1t_1}e^{2kb}\,db = 0. \qquad (39)$$

Further it will be shown that the variables in Eqs. (38), (39) have been chosen so
that they should be easily reduced (under particular assumptions) to the classical
NLS equation.
398   The explicit form of the function $\psi_{1t_1}$ is found by integrating Eq. (22):

$$\psi_{1t_1} = -k\omega|A|^2 e^{2kb} - \int_{-\infty}^b\Omega_2\left(a_2,b'\right)db' - U\left(a_2,t_1\right), \qquad (40)$$


This expression includes three terms. All of them describe a certain component of
the average current. The first one is proportional to the square of the amplitude
modulus and describes the classical potential drift of fluid particles (see
(Henderson et al. (1999) for example). The second one is caused by the presence of
low vorticity in the fluid. Finally, the third item, including $U(a_2,t_1)$, describes an
additional potential flow. It appears in the integration of Eq. (22) over the vertical
coordinate $b$ and will evidently not disappear in case of $A=0$ either. This is a
certain external flow which is chosen depending on a specific problem. Note that a
term of that kind arises in the Eulerian description of potential wave oscillations of
the free surface as well. In the paper by Stocker and Peregrine (1999),
$U = U_* \sin(kx - \omega t)$ was chosen and interpreted as a harmonically changing
surface current induced by an internal wave. We shall further take $U = 0$.
After the substitution of Eq. (40), Eq. (39) may be written in the final form

$$
i\frac{\partial A}{\partial a_2} - \frac{k}{\omega^2}\frac{\partial^2 A}{\partial t_1^2} - k\left( k^2|A|^2 + \beta(a_2) \right)A = 0,
$$

415                                                                                           (41)

$$
\beta(a_2) = \frac{4k^2}{\omega} \int_{-\infty}^{0} e^{2kb}\left( \int_{-\infty}^{b} \Omega_2(a_2,b')db' \right)db.
$$


It is the nonlinear Schrödinger equation for the packet of surface gravity waves
propagating in the fluid with vorticity distribution $\Omega = \varepsilon^2 \Omega_2(a_2,b)$. The function
$\Omega_2(a_2,b)$ determining flow vorticity may be an arbitrary function setting the initial
distribution of vorticity. On integrating it twice we find the vortex component of
the average current which is in no way related to the average current induced by
the potential wave.

## 4 Examples of the waves

Let us consider some special cases following from Eq. (41).

### 4.1 Potential waves

In this case $\Omega_2 = 0$ and Eq. (41) becomes the classical nonlinear Schrödinger
equation for waves in deep water. Three kinds of analytical solutions of the NLS
equation are usually discussed regarding water waves. The first one is the
Peregrine breather propagated in space and time (Peregrine, 1983). This wave may
be considered as a long wave limit of a breather – a pulsating mode of infinite
wavelength (Grimshaw et al., 2010). Two other ones are the Akhmediev breather –
the solution periodic in space and localized in time (Akhmediev et al., 1985) and
the Kuznetsov-Ma breather – the solution periodic in time and localized in space
(Kuznetsov, 1977; Ma, 1979). Both latter solutions evolve against the background
of an unperturbed sine wave.

## 4.2 Gerstner wave

The exact Gerstner solution in complex form is written as (Lamb, 1932; Abrashkin and Yakubovich, 2006; Bennett, 2006):

$$W = a + ib + iA \exp\left[i(ka + \omega t) + kb\right]. \qquad (42)$$

It describes a stationary traveling rotational wave with a trochoidal profile. Its dispersion characteristic coincides with the dispersion of linear waves in deep water $\omega^2 = gk$. The fluid particles are moving in circles and there is no drift current.

Equation (42) is the exact solution of the problem. Following Eqs. (8) and (9) the Gerstner wave should be written as

$$W = a_0 + ib + \sum_{n \geq 1} \varepsilon^n \cdot iA \exp\left[i\left(ka_0 + \omega t_0\right) + kb\right]. \qquad (43)$$

All the functions $w_n$ in Eqs. (8), (9) have the same form. To derive the vorticity of the Gerstner wave, Eq. (43) should be substituted into Eq. (6). Then one can find that in the linear approximation the Gerstner wave is potential ($\Omega_1 = 0$), but in the quadratic approximation it possesses vorticity:

$$\Omega_{2Gerstner} = -2\omega k^2 |A|^2 e^{2kb}. \qquad (44)$$

For this type of the vorticity distribution, the sum of the first two terms in the parentheses in Eq. (41) is equal to zero. From the physical point of view this is due to the fact that the average current induced by the vorticity compensates the potential drift exactly. The packet of weakly nonlinear Gerstner waves in this approximation is not affected by their non-linearity, and the effect of the modulation instability for the Gerstner wave does not occur.

Generally speaking this result is quite obvious. As there is no particle drift in the Gerstner wave, the function $\psi_1$ equals zero. So, the multiplier of the wave amplitude in Eqs. (38), (39) may be neglected without finding vorticity of the Gerster wave.

Let us consider some particular consequences of the obtained result. For the irrotational ($\Omega_2 = 0$) stationary ($A = |A| = const$) wave, Eq. (40) for the velocity of the drifting flow takes on the form

$$\psi_{1t_1} = -\omega k A^2 e^{2kb}. \qquad (45)$$

It coincides with the expression for the Stokes drift in Lagrangian coordinates (in
the Eulerian variables the profile of the Stokes current may be obtained by the
substitution of $b$ for $y$). Thus, our result may be interpreted as a compensation of
the Stokes drift by the shear flow induced by the Gerstner wave in a quadratic
approximation. This conclusion is also fair in the "differential" formulation for
vorticities. From Eq. (22) it follows that the vorticity of the Stokes drift equals the
vorticity of the Gerstner wave with the inverse sign.
The absence of a nonlinear term in the NLS equation for the Gerstner waves
obtained here in the Lagrangian formulation is a robust result and should appear in
the Euler description as well. This follows from the famous Lighthill criterion for
the modulation instability because the dispersion relation for the Gerstner wave is
linear and does not include terms proportional to the wave amplitude.
## 4.3 Gouyon waves
As was shown by Dubreil - Jacotin (1934), the Gerstner wave is a special case of a
wide class of stationary waves having vorticity $\Omega = \varepsilon\Omega_*(\psi)$, where $\Omega_*$ is an
arbitrary function, and $\psi$ is a stream function. Those results were later developed
by Gouyon (1958) who explicitly represented the vorticity in the form of a power
series $\Omega = \sum_{n=1}^{\infty} \varepsilon^n \Omega_n(\psi)$ (see also the monograph by Sretensky (1977)).
When a plane steady flow is considered in the Lagrangian variables, the
stream lines $\psi$ coincide with the isolines of the Lagrangian vertical coordinate $b$
(Abrashkin and Yakubovich, 2006; Bennett, 2006). We are going to consider a
steady-state wave at the surface of an indefinitely deep water. Assume that there is
no undisturbed shear current, but the wave disturbances have vorticity. Then, the
formula for the vorticity is written as $\Omega = \sum_{n=1}^{\infty} \varepsilon^n \Omega_n(b)$. Here we will refer to the
steady-state waves propagating in such a low-vorticity fluid as to the Gouyon
waves. The properties of the Gouyon wave for the first two approximations were
studied by Abrashkin and Zen'kovich (1990) in the Lagrangian description.
In our case, $\Omega_1 = 0, \Omega_2 \neq 0$ and assuming the function $\Omega_2$ to be independent
of the coordinate $a$ we can describe the Gouyon waves. The vorticity $\Omega_2$ depends
on the coordinate $b$ only and has the following form

$$\Omega_{2Goyuon} = \omega k^2 |A|^2 H(kb), \qquad (46)$$

where $H(kb)$ is an arbitrary function. In case of $H(kb) = -2\exp(2kb)$, the vorticities
of the Gerstner and Gouyon waves in the quadratic approximation coincide
(compare Eqs. (44) and (46)). In the considered approximation the Gouyon wave
generalizes the Gerstner wave. From Eq. (22) it follows that the function $\psi_{t_1}$ is
equal to zero only when the vorticity of the Gouyon wave is equal to the vorticity
of the Gerstner wave. Except for this case, the average current $\psi_{t_1}$ will be always
present in the modulated Gouyon waves.
522       The substitution of the ratio (46) into Eq. (41) yields the NLS equation for
the modulated Gouyon wave:

$$i\frac{\partial A}{\partial a_2} - \frac{k}{\omega^2}\frac{\partial^2 A}{\partial t_1^2} - \beta_G k^3 |A|^2 A = 0;$$

525                                                                                     (47)

$$\beta_G = 1 + 4\int_{-\infty}^{0} e^{2\tilde{b}}\left(\int_{-\infty}^{\tilde{b}} H(\tilde{b}')d\tilde{b}'\right)d\tilde{b}; \quad \tilde{b} = kb,$$


where $\tilde{b}$ is a dimensionless vertical coordinate. The coefficient of the nonlinear
term in the NLS equation varies when the wave vorticity is taken into account. For
the Gerstner wave it may be equal to zero like for the Gouyon wave when the
condition


533                                                                                     (48)

$$\int_{-\infty}^{0} e^{2\tilde{b}}\left(\int_{-\infty}^{\tilde{b}} H(\tilde{b}')d\tilde{b}'\right)d\tilde{b} = -\frac{1}{4}.$$



is satisfied. Obviously, an infinite number of distributions of the vorticity $H(\tilde{b})$
meeting this condition are possible. However, realization of one of them seems to
be hardly probable. In the real ocean, distributions of the vorticity with a certain
sign of $\beta_G$ are more likely to be implemented. Its negative values correspond to
the defocusing NLS equation and the positive ones are related to the focusing NLS
equation. In the latter case, the maximum value of the increment as well as the
width of the modulation instability zone of a uniform train of vortex waves vary
depending on the value of $\beta_G$.
544       Equations (39) and (47) will be focusing for $\psi_{1t_1} < 0,\ b \leq 0$ and defocusing if
$\psi_{1t_1} > 0,\ b \leq 0$. The case of the sign-variable function $\psi_{1t_1}$ requires an additional
research. From the physical viewpoint the sign of this function is defined by the
ratio of the velocity of the Stokes drift (45) to the velocity of the current induced
by the vorticity (the integral term in Eq. (40)). For $\psi_{1t_1} < 0$, the Stokes drift either
dominates over a vortex current or both of them have the same direction. When
$\psi_{1t_1} > 0$, the vortex current dominates over the counter Stokes drift. In case of the
sign-variable $\psi_{1t_1}$, the ratio of these currents varies at different vertical levels,
thereby requiring direct calculation of $\beta_G$.

**4.4 Waves with inhomogeneous vorticity distribution along both coordinates**
Neither a vorticity expression nor methods of its definition were discussed when
deriving the NLS equation. In Sections 4.2 and 4.3 devoted to the problems of the
Gerstner and Gouyon waves the vorticity was set to be proportional to a square
modulus of the wave amplitude. Note that waves can propagate against the
background of some vortex current, for example, the localized vortex. In this case
the vorticity may be presented in the form

$$\Omega_2(a_2,b) = \omega\left[\varphi_v(a_2,b) + k^2|A|^2\varphi_w(a_2,b)\right],$$

where the function $\omega\varphi_v$ defines the vorticity of the background vortex current and
the function $\omega k^2|A|^2\varphi_w$ defines the vorticity of waves. In the most general case
both functions depend on the horizontal Lagrangian coordinate as well. Then,
Eq.(41) takes a form

$$i\frac{\partial A}{\partial a_2} - \frac{k}{\omega^2}\frac{\partial^2 A}{\partial t_1^2} - k\beta_v(a_2)A - k^3\left(1 + \beta_w(a_2)\right)|A|^2 A = 0,$$

570 (49)

$$\beta_{v,w}(a_2) = 4\int_{-\infty}^{0} e^{2\tilde{b}}\left(\int_{-\infty}^{\tilde{b}}\varphi_{v,w}(a_2,\tilde{b}')d\tilde{b}'\right)d\tilde{b}.$$

The substitution

$$A_* = A\exp\left(-ik\int_{-\infty}^{a_2}\beta_v(a_2)da_2\right)$$
(50)

reduces Eq. (49) to the NLS equation with a non-uniform multiplier for the
nonlinear term:

$$i\frac{\partial A^*}{\partial a_2} - \frac{k}{\omega^2}\frac{\partial^2 A^*}{\partial t_1^2} - k^3\left(1 + \beta_w(a_2)\right)\left|A^*\right|^2 A^* = 0.$$
(51)

Let us consider the propagation of the Gouyon wave, when $\beta_w = const = \beta_G - 1$
and Eq.(51) turns into the classical NLS equation (47). As shown in Sec. 4.3, it
describes the modulated Gouyon waves. Therefore, on the substitution of Eq. (50)
one can conclude that the propagation of the Gouyon waves against the
background of the non-uniform vortex current results in the variation of the wave
number of the carrier wave. For $\beta_w = 0$, Eq. (51) describes the propagation of a
packet of potential waves against the background of the non-uniform weakly
vortical current. The specific features of the wave propagation related to the
variable $\beta_w$ require special investigation.

**5 On equivalence of Lagrangian and Eulerian approaches**

Consider the correlation between the Eulerian and the Lagrangian description of wave packets. To obtain the value for elevation of the free surface we substitute the expressions (8), (9), (13) and $b = 0$ into the equation for $Y = \text{Im}\,W$ written in the following form

$$Y_L = \varepsilon\,\text{Im}\,A(a_2, t_1)\exp i(ka_0 + \omega\,t_0),$$

where $A(a_2, t_1)$ is the solution of Eq. (41). This expression defines the wave profile in Lagrangian coordinates. To rewrite this equation in the Eulerian variables it is necessary to define $a$ via $X$. From the relation (8) follows

$$X = a + \varepsilon\,\text{Re}\left(w_1 + \sum_{n=2}\varepsilon^{n-1}w_n\right) = a + O(\varepsilon),$$

and the elevation of the free surface in the Eulerian variables $Y_E$ will be written as

$$Y_E = \varepsilon\,\text{Im}\,A(X_2, t_1)\exp i(kX_0 + \omega\,t_0) + O(\varepsilon^2); \quad X_l = \varepsilon^l X.$$

The coordinate $a$ plays the role of $X$, so the following substitutions are valid for the Lagrangian approach:

$$a_0 \to X_0; \quad a_1 \to X_1; \quad a_2 \to X_2.$$

This result may be called an "equivalence principle" between the Lagrange and the Euler descriptions for solutions in the linear approximation. This principle is valid for both the potential and rotational waves.

To express the solution of Eq. (41) in the Eulerian variables it is necessary to use the equivalence principle and to replace the horizontal Lagrangian coordinate $a_2$ by the $X_2$ coordinate. So, there are no discrepancies between the Eulerian and the Lagrangian estimations of the NLS equation for the free surface elevation.

Taking this into account we can conclude that the result will be the same in the Eulerian description, if the vorticity $\Omega_2$ is a function of the $x, y$ coordinates. So, when studying the wave packets dynamics in the vortical liquid in the Eulerian variables it is necessary to replace (ex. in Eq. (41) or (51)) the horizontal Lagrangian coordinate by the Eulerian one.

Equation (47) can also be derived in Eulerian variables. The key idea is to take into consideration a weak shear flow. This approach is similar to the method used in the paper by Trulsden and Hejervick (2009), where the wave propagates along a weak horizontal shear current. Shrira and Slunyaev (2014) used this technique to study trapped waves in an uniform jet stream. They derived the NLS

equation for a single mode. Later, Slunyaev (2016) generalized the result to the case of a vortex jet flow. Our result was obtained with a weak vertical shear flow taking into account. In particular, to describe modulated Guyon waves, the Johnson approach (1976) should be modified, assuming a shear flow of the order of epsilon.

The solutions of the considered problem in the Lagrange and the Euler forms in the quadratic and cubic approximations differ from each other. To obtain a full solution in the Lagrange form one should find functions $\psi_1, \psi_2, \psi_3, f_2, f_3$. This problem should be considered within a special study.

## 6 Conclusion

We have derived the vortex-modified nonlinear Schrödinger equation using the method of multiple scale expansions in the Lagrange variables. The fluid vorticity $\Omega$ is specified as an arbitrary function of the Lagrangian coordinates, which is quadratic in the small parameter of the wave steepness. The calculations have been performed introducing a complex coordinate of the fluid particle trajectory.

The nonlinear evolution equation for the wave packet in the form of the nonlinear Schrödinger equation has been derived as well. From the mathematical viewpoint, the novelty of this equation is related to the emergence of a new term proportional to the envelope amplitude and the variance of the coefficient of the nonlinear term. If the vorticity depends on the vertical Lagrangian coordinate only (the Gouyon waves), this coefficient is constant. There are special cases, when the coefficient of the nonlinear term equals zero and the resulting non-linearity disappears. The Gerstner wave belongs to the latter case. Another effect revealed in the present study is the relation of the vorticity to the wave number shift in the carrier wave. This shift is constant for the modulated Gouyon wave. If the vorticity depends on both Lagrangian coordinates, the shift of the wave number is horizontally inhomogeneous. It is shown that the solution of the NLS equation for weakly rotational waves in the Eulerian variables may be obtained from the Lagrangian solution by an ordinary change of the horizontal coordinates.

*Acknowledgments.* EP appreciates the support obtained from the RNF under grant 16-17-00041. Authors wish to thanks the Editor, Roger Grimshaw and reviewers for very useful comments, and Nadezhda Krivatkina for English corrections.

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
