# Peer review of "Lagrange form of the nonlinear Schrödinger equation for"

_Nonlinear Processes in Geophysics, 2016_

## Referee Comment (RC1) · Anonymous Referee #1 · 12 Jan 2017

*"The Lagrange form of the nonlinear Schrödinger equation for low vorticity waves in deep water: rogue wave aspect"*

**by A.Abrashkin & E.Pelinovsky**

The manuscript is devoted to a well studied aspect of water wave dynamics - weakly nonlinear evolution of a narrow band wavetrain. The work contains two novel elements: the focus of the consideration is on waves with weak vorticity, which has not been a subject of a dedicated study before; the analysis and results are formulated in terms of Lagrangian variables. The main result is the derivation of the nonlinear Schrödinger equation for the wavetrain envelope in the Lagrangian variables and an analysis of a few examples on its basis. The derivation is sound. The results are discussed from the rogue wave perspective. Although the work does present new results, seems to be correct and is clearly written, I cannot recommend its publication yet. However, it has a potential to be turned into a much better paper. To this end a revision is needed. The specific points to address are outlined below.

**The main points:**

(i) **The motivation:** From the provided literature review it is not clear why this particular study is needed: ***What are the specific questions that the authors want to clarify? Why these questions might be of interest and for what segments of the scholar community?*** Wavetrain modulations upon arbitrary vertically sheared currents were thoroughly studied by Benney and his group. If the Benney asymptotic expansion becomes invalid for the range of small values of vorticity the present work is focussing upon, then it has to be shown and explained what is wrong with the Benney expansion. The same question applies to Jonsson (1976) results.

The dependence of the cubic nonlinearity on vorticity in Jonsson (1976) and the works by Benney is not singular. Therefore similar expansion for the small vorticity can be carried out in the Eulerian framework as well using the known results, say, by Jonsson (1976) and/or the works by Benney group as the starting point. This should be made clear. I think what the authors are doing is a re-derivation of the NLS for weak vorticity; the results were known, although implicitly, since nobody looked specifically into this case. Hence there is indeed a novelty here, but a comparison with the Eulerian results is necessary. In the Eulerian case vorticity can also be always presented as an expansion in $\varepsilon$, although in contrast with the Lagrangian approach only the leading order vorticity will be constant. In this context the most intriguing question is concerned with one of the highlights of the work: the vanishing of the cubic nonlinearity in the NLS in the Lagrangian variables for the Gerstner wave. (This result is more significant than the authors give it credit for: it shows that in principle an $O(\varepsilon^2)$ shear might kill the NLS nonlinearity.) The question is: what is the manifold of Eulerian shear profiles (or vorticity distributions) which would zero the NLS nonlinearity? I believe it could be answered by a straightforward analysis of the known expressions for the coefficient. Also the similar question applies to the Lagrangian formulation: the vorticity distribution is arbitrary, what are other

distributions for which the NLS nonlinearity vanishes? I doubt that the Gerstner is an isolated special case.

It follows from the works by Benney and his group that the transverse instability is much stronger than the longitudional one, therefore, the studies of strictly longitudional instabilities are of limited interest from the viewpoint of sea applications and could be applied only to narrow wave tanks. I'd like this point to be mentioned more explicitly in the introduction. This is important since it squarely places the derived NLS into the realm of toy models. This does not mean that the results cannot be of interest or should not be published, it just means that the results might interest a different community.

The original element of the work is the asymptotic derivation of the NLS in Lagrangian variables. In my view this is complementary to the existing Eulerian works and it remains unclear what new features/aspects this might reveal.

(ii) **The NLS**: In contrast to the NLS in Euler variables where we know that the equation describes evolution of the envelope amplitude in the $(x, t)$ space and how the actual elevation can be expressed as a Stokes-like series in wave amplitude up to cubic order, here the NLS in Lagrangian variables is an object which is much less straightforward to interpret. Obviously, $A$ is the envelope amplitude, but what are the independent variables $(a, b)$? Their link to the standard Eulerian variables $(x, y)$ is not known. Although it is straightforward, at least in principle, to provide this link in terms of a series in $\varepsilon$, the authors choose not to do this. They effectively use the zero order approximation where the difference between the Eulerian and Lagrangian descriptions vanishes. Then the rationale for using the Lagrangian approach apparently disappears.

I suspect (this is the most interesting point), that if the authors make transformation to return to the Euler variables, they will get a higher order NLS type equation since the transformation itself is nonlinear (see e.g. F.Nouguier, B. Chapron, C-A, Guйrin *Second-order Lagrangian description of tri-dimensional gravity wave interactions*, JFM 772, 165-196, (2015) and references therein).

If the authors do not want to go through this straightforward but quite time consuming path I suggested above, then they can handle the comparison numerically. The Lagrangian solution yields $X, Y$ in terms of $a, b, t$. Hence the surface elevation $Y(a, 0, t)$ and position of a parcel on the free surface, $X(a, 0, t)$, which are found in terms of a series, provide implicit function $Y(a, 0, t)$ which can be easily plotted for a typical $Y(X, t)$, say, a breather. This plot has to compared to the Eulerian solution with the cubic terms retained.

The obtained NLS is presented in an "optical"form (with space rather than time chosen as the propagation variable), which is a somewhat strange choice for a

hydrodynamic work. Dependence on $t$ in this context means dependence on the running variable. I do not understand why the authors choose this form and stick to it, they give no clue. They have either argue for their preference or switch to the conventional form.

The authors consider the NLS derivation allowing for horizontal nonuniformity, which raises a host of questions. How arbitrary the dependence on $a_2$ is? What does it mean? Are the $a_2$ dependencies of this vortical and potential parts of the Doppler correction linked to satisfy the Lagrange equations? How these dependencies can be specified?

**(iii) Rogue waves:** As I've already mentioned, the strong transverse instability of the wavetrains does not allow one to speak seriously about ocean applications. I found nothing new and specific adding to our understanding of rogue waves. The fact that the NLS is formulated in the Lagrangian variables and only the leading order term is used makes this equation equivalent (to this order) to the Eulerian NLS. The fact that in the focussing NLS there is modulational instability and that such a NLS admits breather solutions is known for about thirty years.

The term "rogue wave"is used in the manuscript as synonymous with the term breather, just because the latter satisfy the rogue wave amplitude criterion. Although the NLS breather solutions are indeed often used as prototypes of rogue waves, this could be done only with appropriate explicitly spelled out caveats.

The weakest point in the rogue wave aspect of the paper is that I don't see any new insight into the nature of rogue waves even in the framework of the chosen toy model.

In my view the following question might be of interest in the context of rogue waves and would have an element of novelty: what is the profile and maximal height of the found Akhmediev Lagrangian breather in the Eulerian variables. To answer this question the authors have to sum up all orders of their expansion and then perform the transformation to the Eulerian variables. The results will differ from the corresponding expansion in the Eulerian variables. I re-iterate that it would be of interest to discuss this difference. I've mentioned already the simplest way to get it.

---

## Referee Comment (RC2) · Anonymous Referee #2 · 9 Feb 2017

The paper describes a new derivation of the NLS equation, based on a Lagrangian coordinates approach, in the presence of weak vorticity. First, an introduction presents several previously existing derivations of the NLS equation, and offers an interesting review of recent developments designed to take vorticity into account. Then, the Lagrange coordinates, and associated general equations are presented in section 2, while the new NLS equation related to this framework is derived in section 3. Several results are presented at the end of section 3, and in section 4 (only those related to envelope soliton solutions), and summarized in section 5. The paper is relatively well structured, even if several typos remain. Globally, several new results can be found in the manuscript, and for all these reasons, I recommend publication, after some modifi-

cations, in "Nonlinear Process in Geophysics".

Still, several concerns remain. Addressing them could help improving the manuscript.

- First, it suffers a lack of illustration. Indeed, a single figure appears, and intends to show the full geometry of the problem. For instance, from this figure, I cannot understand what is this "average current" (average in time, in 'a' coordinate? In 'b' coordinate?). Neither can I see a weak vorticity. Thus, the definition of vorticity is confusing. Another way to say the same thing is that the Euler to Lagrange coordinate transform is not clear. Is a background vortical flow included? Or do we only consider the vortical flow induced by the waves?

- Presentation of the results is a little bit confusing.

o For instance, it is shown that in the absence of vorticity and current, the Akhmediev soliton solution in Lagrange coordinates does correspond to the Akhmediev soliton in Euler coordinates, up to the second order in epsilon. But then, for quadratic and cubic terms, it is claimed the solutions differ. Here could be an interesting result. Could the authors consider obtaining these solutions, and present the differences?

o When considering the Gerstner wave, where does the vorticity profile comes from? Thus, the following sentence is disturbing: "From the physical point of view, this is due to the fact that the average current induced by vorticity exactly compensates the stokes drift". Is it only true when integrated? The result associated is very interesting (finding Gerstner waves not affected by modulational instability), but its explanation is not straightforward, and should be developed. Still, these waves are a very specific case, and this is not clear from the text.

o Results of the following part, entitled "Rogue waves", describe the evolution of the coefficients of the NLS equation with the structure of vorticity. In each one of the three cases studied, the eventuality of a Peregrine breather soliton to exist is analyzed. Maybe, the characteristics of these new Peregrine breather could be described, by

comparison with classical one. This analysis would provide an idea of whether or not a vortical flow is amenable to increase the probability of occurrence of rogue waves.

In conclusion, I consider the paper to be good, and to bring several new results. It is well structured, and the results are clearly new. However, the authors might develop several points, and improve the impact of this work.

─────────────────────────────

---

## Author Comment (AC1)

**Answers on review's comments on paper**

**The Lagrange form of the nonlinear Schrodinger equation for low vorticity waves in deep water: rogue wave aspect**

*by*

*Anatoly Abrashkin and Efim Pelinovsky*

**RESPONSE TO REVIEWER 1**

**THE MOTIVATION:**

**Review 1:**

From the provided literature review it is not clear why this particular study is needed? *What the specific questions that the authors want to clarify? Why these questions might be of interest and for what segments of the scholar community?*

**Authors:**

**We found a new family of solutions** for the wave train propagation in the deep water. Their novelty is **non-uniform distribution of the vorticity**.

**Reviewer 1:**

Wavetrain modulations upon arbitrary vertically sheared currents were thoroughly studied by Benny and his group. If the Benny asymptotic expansion becomes invalid for the range of small values of vorticity the present work is focusing upon, then it has to be shown and explained what is wrong with the Benny expansion. The same question applies to Jonhson (1976) results. The dependence of the cubic nonlinearity on vorticity in Jonhson (1976) and the works by Benny is not singular. Therefore similar expansion for the small vorticity can be carried out in the Eulerian framework as well using the known results, say, by Jonhson (1976) and/or the works by Benny group as the starting point. I think what the authors are doing is a re-derivation of the NLS for weak vorticity; the results were known, although implicitly, since nobody looked specifically into this case.

**Authors:**

We study flows with the vorticity depending on both Lagrange coordinates. That corresponds to the background current depending on the variables $x, y, t$ in the Eulerian approach, not a shear flow $U(y)$ as Johnson or Benny studies. **Our approach differs from other known ones cardinally.**

**Reviewer 1:**

Hence there is indeed a novelty here, but a comparison with the Eulerian results is necessary. In the Eulerian case vorticity can also be always presented as an explanation in *epsilon*, although in contrast with Lagrangian approach only the leading order vorticity will be constant.

**Authors:** Yes! But the functions of vorticity's row $\Omega_n$ ($n > 1$) depend on $x, y, t$, i.e. $\Omega_n = \Omega_n(x, y, t)$. They are not integrals of motion as in the Lagrangian description. So our question to the reviewer: how to set these functions? It is obvious that **the Lagrange approach is more preferable in that situation.**

**Reviewer 1:**

In this context the most intriguing question is concerned with one of highlights of the work: the vanishing of the cubic nonlinearity in the NLS in the Lagrangian variables for the Gerstner wave. (This result is more significant than the authors it credit for: it shows that in principle an $O(\varepsilon^2)$ shear might kill the NLS nonlinearity. The question is: what is the manifold of Eulerian shear profiles (or vorticity distributions which would zero the NLS nonlinearity? I believe it could be answered by a straightforward analysis of the known expressions for the coefficient.

**Authors**: **That's right.** Using the accordance principle (pages 13, 14) we can conclude that the shear flow with the vorticity equal to the vorticity of the Gerstner wave kills the nonlinearity in the NLS equation.

**Reviewer 1:** Also the similar question applies to the Lagrangian formulation: the vorticity distribution is arbitrary, what are other distributions for which the NLS nonlinearity vanishes? I doubt that the Gerstner is an isolated special case.

**Authors**: The NLS equation's nonlinearity vanishes if $\psi_{1t_1} = 0$. The Gerstner wave is **the single solution** of this equation.

**Reviewer 1:** It follows from the works by Benny and his group that transverse instability is much stronger than the longitudinal one, therefore, the studies of strictly longitudinal instabilities are limited interest from the viewpoint of sea applications and could be applied only to narrow wave tanks. I'd like this point to be mentioned more explicitly in the introduction. This is important since it squarely places the derived NLS into the realm of toy models. This does not mean that the results cannot be of interest or should not be published, it just means that the results might interest a different community.

**Authors**: We mentioned Benny's result in the introduction (lines 63-68). And suppose that it is quite enough. In our opinion the limit of longitudinal instability is

rather applicable for open sea conditions where the length to width ratio of a non-linear wave package is much less than in any restricted waters.

As for narrow tanks they are usually much more active in view of transverse instability and need special tuning to avoid this effect. The reviewer is absolutely correct in this state. So the general formulation of the model under the condition of the rigid borders could be applicable for interpretation of experimental results as well.

**Reviewer 1:** The original element of the work is the asymptotic derivation of the NLS in Lagrangian variables. In my view this is complementary to the existing Eulerian works and it remains unclear what new features/aspects this might reveal.

**Authors**: The original aspect of our study is **horizontal non-uniformity of vorticity's distribution** (lines 126-128). As a consequence, in contrast to Benny and his group and Johnson we derived the evolutional equation with variable coefficients.
**THE NLS:**

**Reviewer 1:** In contrast to the NLS in Euler variables where we know that the equation describes evolution of the envelope amplitude in the $(x, t)$ space and how the actual elevation can be expressed as a Stokes-like series in wave amplitude up to cubic order, here the NLS in Lagrangian variables is an object which is much less straightforward to interpret. Obviously, $A$ is the envelope amplitude, but what are the independent variables $(a, b)$?

**Authors**: Lagrangian variables are the labels of the fluid particles, nothing more over.

**Reviewer 1:** Their link to the standard Eulerian variables $(x, y)$ is not known. Although, it is straightforward, at least in principle, to provide this link in terms of series in $\varepsilon$, the authors choose not to do this. The effectively use the zero order approximation where the difference between the Eulerian and Lagrangian description vanishes. Then the rationale for using the Lagrangian approach apparently disappears.

**Authors**: **That is not correct.** We derived a new family of solutions due to Lagrange approach. It is much more difficult problem to get them in Eulerian description which has not been solved yet.

**Reviewer 1:** I suspect (this is the most interesting point), that if the authors make transformation to return to the Euler variables, they will get a higher order NLS type equation since the transformation itself is nonlinear (see e.g. F. Nouguier, B. Chapron, C-A, Guérin Second-order Lagrangian description of tri-dimensional gravity wave interactions, JFM 772, 165-196 (2015) and references therein).

**Authors**: That is a special problem. We are ready to discuss it further.

**Reviewer 1:** If the authors do not want to go through this straightforward but quite time consuming pass I suggested above, then they can handle the comparison numerically. The Lagrangian solution yields $X, Y$ in terms of $a, b, t$. Hence the surface elevation $Y(a,0,t)$ and position of a parcel on the free surface, $X(a,0,t)$, which are found in terms of series, provide implicit function $Y(a,0,t)$ which can be easily plotted for a typical $Y(X,t)$, say, a breather. This plot has to compared to the Eulerian solution with the cubic terms retained.

**Authors**: That is a good programme, but nobody has calculated the Eulerian solution with the cubic terms. All authors are restricted to the derivation of the NLS equation. With what solution do we have to compare our results? Or we must study our problem in Euler variables too? Besides, we are interested in rogue waves in this paper and study the leading order of the solution only. The terms of the second and cubic orders are out of our attention.

**Reviewer 1:** The obtained NLS is presented in an "optical" form (with space rather than time chosen as the propagation variable), which is somewhat strange choice for a hydrodynamic work. Dependence on $t$ in this context means dependence on running variable. I do not understand why the authors choose this form and stick to it, they give no clue. They have either argue for their preference or switch to the conventional form.

**Authors**: In traditional hydrodynamics form (in variables $a_1, a_2, t_2$) it is impossible to lead our evolutional equation to the usual NLS equation. So it is used the "optical" variant of the equation. We shall switch that explanation in the conventional form.

**Reviewer 1:** The authors consider the NLS derivation allowing for horizontal non-uniformity, which raises a host of questions. How arbitrary the dependence on $a_2$ is? What does it mean? Are the $a_2$ dependencies of these vertical and potential parts of the Doppler correction linked to satisfy the Lagrange equations? How these dependencies can be specified?

**Authors**: The vorticity $\Omega_2(a_2, b)$ is arbitrary continuous differentiable bounded function. That means the boundedness of the vorticity or the derivatives of the field of the velocity. The vertical and horizontal parts of the Doppler correction don't link. It is obvious from the comparison of the equations (41) and (44).

**ROGUE WAVES:**

**Reviewer 1:** As I've already mentioned, the strong transverse instability of the wavetrains does not allow one to speak seriously about ocean applications. I found nothing new and specific adding to our understanding of rogue waves. The fact that the NLS is formulated in the Lagrangian variables and only the leading order term is used makes this equation equivalent (to this order) to the Eulerian NLS. The fact that in the focusing NLS there is modulational instability and that such NLS admits breather solutions is known for about thirty years.

**Authors**: Definitely we do not propose a new understanding of rogue wave formation, just the original approach of their description. And in this sense our solutions are not analogous to any known ones.

**Reviewer 1:** The term "rogue wave" is used in the manuscript as synonymous with the term breather, just because the latter satisfy the rogue wave criterion. Although the NLS breather solutions are indeed often used as prototypes of rogue waves, this could be done only with appropriate explicitly spelled out caveats.

**Authors**: The last phrase could be considered as a private opinion of the reviewer. Using the corresponding terms we just follow traditions of the scientific community.

**Reviewer 1:** The weakest point in the rogue wave aspect of the paper is that I don't see any new insight into the nature of rogue waves even in the framework of the chosen toy model.

**Authors**: The novelty of the present paper is that we proved a possibility of formation and propagation of the rogue waves at the background of the horizontally non-uniform current. One can consider a single localized vortex as an example of this solution. The waves of such type didn't study yet.

**Reviewer 1:** In my view the following question might be of interest in the context of rogue waves and would have an element of novelty: what is the profile and maximal height of the found Akhmediev Lagrangian breather in the Eulerian variables. To answer this question the authors have to sum up all orders of their expansion and then perform the transformation to the Eulerian variables. The results will differ from the corresponding expansion in the Eulerian variables. I reiterate that it would be of interest to discuss this difference. I've mentioned already the simplest way to get it.

**Authors**: That is a good idea. But we have stress: where does the reviewer see the explicit solutions up to the third order in the Eulerian variables? And why nobody has accomplished this work in the Eulerian description? We assume that the reviewer doesn't understand the complexity of his suggestion. He proposes a big new project which could be the subject of a new paper.

---

## Author Comment (AC2)

**Answers on the comments of Reviewer 2 of the paper**

**The Lagrange form of the nonlinear Schrodinger equation for low vorticity waves in deep water: rogue wave aspect**
**by**
**Anatoly Abrashkin and Efim Pelinovsky**

**Reviewer 2:**
First, it suffers a lack of illustration. Indeed, a single figure appears, and intends to show the full geometry of the problem. For instance, from the figure, I cannot understand what is this "average current" (average in time, in 'a' coordinate? In 'b' coordinate?) Neither can I see a weak vorticity. Thus, the definition of vorticity is confusing. Another way to say the same thing is that the Euler to Lagrange coordinate transform is not clear. Is a background vertical flow included? Or do we only consider the vertical flow induced by the waves?

**Authors:**
We highly appreciate this comment. We shall add the required information in the figure. The horizontal current $\psi_{1t_1}$ depends on slow coordinates only. So we name it "average current" as it is average in fast variables $a_0, t_0$ (see Eq.13). A weak vorticity is set as an arbitrary function of the Lagrange coordinates in some region. For example, the vorticity may differ from zero within the restricted region. This case corresponds to the interaction of the wave with the localized vortex. We plan to illustrate this distribution of the vorticity in the figure. We concentrate on studying horizontal current because it is a term of the NLS equation. But in the quadratic approximation there exists the vertical flow as well. This one is described by the function $f_2$ depending on the wave amplitude and the horizontal current (see Eq.21). In the Eulerian coordinates the background flow has two components of the velocity depending on the variables $x, y, t$.

**Reviewer 2:**
Presentation of the results is a little bit confusing.
For instance, it is shown that the absence of vorticity and current, the Akhmediev soliton solution in Lagrange coordinates does correspond to the Akhmediev soliton in Euler coordinates, up to the second order in epsilon. But then, for quadratic and cubic terms, it is claimed the solutions differ. Here could be an interesting result. Could be the authors consider obtaining these solutions and present differences?

**Authors:**
That is a good idea, but there are two serious problems. First, it is necessary to get a solution up to the third order in the Euler coordinates. As far as we know nobody was able to do that. Second, it is necessary to transform our solution to the Eulerian form, i.e. to solve equations for the functions $\psi_1, \psi_2, \psi_3, f_2, f_3$ and then to express the obtained solutions in the Euler variables. This program requires very unwieldy calculations. So we consider the reviewer's idea worth being realized in a new paper.

**Reviewer 2:**
When considering the Gerstner wave, where does the vorticity profile comes from? Thus, the following sentence is disturbing: "From the physical point of view, this is due to the fact that the average current induced by the vorticity compensates the stokes drift". Is it only true when integrated? The result associated is very interesting (finding Gerstner waves not affected by

modulational instability), but its explanation is not straightforward and should be developed. Still, these waves are a very specific case, and this is not clear from the text.

**Authors:**
We agree that a more detailed explanation is necessary. The vorticity profile of the Gerstner wave is found by substitution of the solution (Eq. 42) into Eq. 6. This will be mentioned in the revised paper. Besides, special attention will be paid to the fact that the first term in Eq. 40 coincides with the Stokes drift. It is a remarkable fact. In the quadratic approximation the vorticity of the Gerstner wave equals modulo and opposite in sign to the vorticity of Stokes drift. So these terms mutually neglect each other without integration (see Eq. 22). The Gerstner wave is really a very specific case and we shall mention this fact in the revised paper.

**Reviewer 2:**
Results of the following part, entitled "Rogue waves", describe the evolution of the coefficients of the NLS equation with the structure of vorticity. In each one of the three cases studied, the eventuality of a breather soliton to exist is analyzed. Maybe, the characteristics of these new Peregrine breather could be described, by comparison with classical one. This analysis would provide an idea of whether or not a vertical flow is amenable to increase the probability of occurrence of rogue waves.

**Authors:** The vorticity leads to variation of the wavelength of the carrier wave but doesn't affect its amplitude. The vertical flow is described by the function $f_2$ (see Eq. 21). It affects the amplitude in the next approximation relatively to the solutions of the NLS equation.

**The authors would like to express their sincere thanks to Reviewer 2 for all of the valuable comments and helpful suggestions.**

---

## Author Response (AR1)

**Dear Professor Grimshaw!**
**We revised our manuscript according to the comments of the Reviewers which are definitely useful for the paper's content and are accepted with our gratitude.**

**Sincerely yours,**
**Anatoly Abrashkin and Efim Pelinovsky**

**Answers on review's comments on paper**

**The Lagrange form of the nonlinear Schrödinger equation for low vorticity waves in deep water: rogue wave aspect**

*by*

*Anatoly Abrashkin and Efim Pelinovsky*

**RESPONSE TO REVIEWER 1**

**THE MOTIVATION:**

**Review 1:**

From the provided literature review it is not clear why this particular study is needed? *What the specific questions that the authors want to clarify? Why these questions might be of interest and for what segments of the scholar community?*

**Authors:**

**We found a new family of solutions** for the wave train propagation in the deep water. Their novelty is **non-uniform distribution of the vorticity**.

**Reviewer 1:**

Wavetrain modulations upon arbitrary vertically sheared currents were thoroughly studied by Benny and his group. If the Benny asymptotic expansion becomes invalid for the range of small values of vorticity the present work is focusing upon, then it has to be shown and explained what is wrong with the Benny expansion. The same question applies to Jonhson (1976) results. The dependence of the cubic nonlinearity on vorticity in Jonhson (1976) and the works by Benny is not singular. Therefore similar expansion for the small vorticity can be carried out in the Eulerian framework as well using the known results, say, by Jonhson (1976) and/or the works by Benny group as the starting point. I think what the authors are doing is a re-derivation of the NLS for weak vorticity; the results were known, although implicitly, since nobody looked specifically into this case. Hence there is indeed a novelty here, but a comparison with the Eulerian results is necessary. In the Eulerian case vorticity can also be always presented as an explanation in *epsilon*, although in contrast with Lagrangian approach only the leading order vorticity will be constant.

**Authors:** We added (Sec. 5, lines 626-630):

Taking this into account one could conclude that the result will be the same in the Eulerian description if the vorticity $\Omega_2$ will be set as a function of the coordinates $x, y$. Respectively when studying dynamics of wave packets in the vortical liquid in the Eulerian variables it is necessary to replace (ex. in Eq. (41) or (51)) the horizontal Lagrangian coordinate by the Eulerian one.

**Reviewer 1:**

In this context the most intriguing question is concerned with one of highlights of the work: the vanishing of the cubic nonlinearity in the NLS in the Lagrangian variables for the Gerstner wave. (This result is more significant than the authors it credit for: it shows that in principle an $O(\varepsilon^2)$ shear might kill the NLS nonlinearity. The question is: what is the manifold of Eulerian shear profiles (or vorticity distributions which would zero the NLS nonlinearity? I believe it could be answered by a straightforward analysis of the known expressions for the coefficient.

**Authors**: We added (Sec. 4.2, lines 477-489):

[revised manuscript text omitted]

**Reviewer 1:** It follows from the works by Benny and his group that transverse instability is much stronger than the longitudinal one, therefore, the studies of strictly longitudinal instabilities are limited interest from the viewpoint of sea applications and could be applied only to narrow wave tanks. I'd like this point to be mentioned more explicitly in the introduction. This is important since it squarely places the derived NLS into the realm of toy models. This does not mean that the results cannot be of interest or should not be published, it just means that the results might interest a different community.

**Authors**: We mentioned Benny's result in the introduction (lines 41-49). And suppose that it is quite enough.

**Reviewer 1:** The original element of the work is the asymptotic derivation of the NLS in Lagrangian variables. In my view this is complementary to the existing Eulerian works and it remains unclear what new features/aspects this might reveal.

**Authors**: The original aspect of our study is **horizontal non-uniformity of vorticity's distribution**. As a consequence, in contrast to Benny and his group and Johnson we derived the evolutional equation with variable coefficients.

**THE NLS:**

**Reviewer 1:** In contrast to the NLS in Euler variables where we know that the equation describes evolution of the envelope amplitude in the $(x,t)$ space and how the actual elevation can be expressed as a Stokes-like series in wave amplitude up to cubic order, here the NLS in Lagrangian variables is an object which is much less straightforward to interpret. Obviously, $A$ is the envelope amplitude, but what are the independent variables $(a,b)$?

**Authors**: Lagrangian variables are the labels of the fluid particles, nothing more over.

**Reviewer 1:** Their link to the standard Eulerian variables $(x,y)$ is not known. Although, it is straightforward, at least in principle, to provide this link in terms of series in $\varepsilon$, the authors choose not to do this. The effectively use the zero order approximation where the difference between the Eulerian and Lagrangian description vanishes. Then the rationale for using the Lagrangian approach apparently disappears.

**Authors**: We derived a new family of solutions due to Lagrange approach. A problem of their Eulerian description has not been solved yet.

**Reviewer 1:** I suspect (this is the most interesting point), that if the authors make transformation to return to the Euler variables, they will get a higher order NLS type equation since the transformation itself is nonlinear (see e.g. F. Nouguier, B. Chapron, C-A, Guérin Second-order Lagrangian description of tri-dimensional gravity wave interactions, JFM 772, 165-196 (2015) and references therein).

**Authors**: That is a special problem. We are ready to discuss it further.

**Reviewer 1:** If the authors do not want to go through this straightforward but quite time consuming pass I suggested above, then they can handle the comparison numerically. The Lagrangian solution yields $X,Y$ in terms of $a,b,t$. Hence the surface elevation $Y(a,0,t)$ and position of a parcel on the free surface, $X(a,0,t)$, which are found in terms of series, provide implicit function $Y(a,0,t)$ which can be easily plotted for a typical $Y(X,t)$, say, a breather. This plot has to compared to the Eulerian solution with the cubic terms retained.

**Authors**: That is a good programme, but nobody has calculated the Eulerian solution with the cubic terms. All authors are restricted to the derivation of the NLS equation. With what solution do we have to compare our results? Or we must study our problem in Euler variables too? Besides, we are interested in rogue

waves in this paper and study the leading order of the solution only. The terms of the second and cubic orders are out of our attention.

**Reviewer 1:** The obtained NLS is presented in an "optical" form (with space rather than time chosen as the propagation variable), which is somewhat strange choice for a hydrodynamic work. Dependence on $t$ in this context means dependence on running variable. I do not understand why the authors choose this form and stick to it, they give no clue. They have either argue for their preference or switch to the conventional form.

**Authors**: We added (Sec. 3.3, after Eq. (39), lines 398-400):

Further it will be shown that variables in Eqs. (38), (39) were chosen in the easiest form for their reduction (under the particular assumptions) to the classical NLS equation.

**Reviewer 1:** The authors consider the NLS derivation allowing for horizontal non-uniformity, which raises a host of questions. How arbitrary the dependence on $a_2$ is? What does it mean? Are the $a_2$ dependencies of these vertical and potential parts of the Doppler correction linked to satisfy the Lagrange equations? How these dependencies can be specified?

**Authors**: The vorticity $\Omega_2(a_2, b)$ is an arbitrary bounded function. The vertical and horizontal parts of the Doppler correction don't link. It is obvious from the comparison of the equations (41) and (44).

**ROGUE WAVES:**

**Reviewer 1:** As I've already mentioned, the strong transverse instability of the wavetrains does not allow one to speak seriously about ocean applications. I found nothing new and specific adding to our understanding of rogue waves. The fact that the NLS is formulated in the Lagrangian variables and only the leading order term is used makes this equation equivalent (to this order) to the Eulerian NLS. The fact that in the focusing NLS there is modulational instability and that such NLS admits breather solutions is known for about thirty years. The term "rogue wave" is used in the manuscript as synonymous with the term breather, just because the latter satisfy the rogue wave criterion. Although the NLS breather solutions are indeed often used as prototypes of rogue waves, this could be done only with appropriate explicitly spelled out caveats. The weakest point in the rogue wave aspect of the paper is that I don't see any new insight into the nature of rogue waves even in the framework of the chosen toy model. In my view the following question might be of interest in the context of rogue waves and would have an

element of novelty: what is the profile and maximal height of the found Akhmediev Lagrangian breather in the Eulerian variables. To answer this question the authors have to sum up all orders of their expansion and then perform the transformation to the Eulerian variables. The results will differ from the corresponding expansion in the Eulerian variables. I re-iterate that it would be of interest to discuss this difference. I've mentioned already the simplest way to get it.

**Authors**: We excluded the discussion of the problem of rogue waves from our paper.

**Answers on the comments of the Reviewer 2 on paper**

**The Lagrange form of the nonlinear Schrödinger equation for low vorticity waves in deep water: rogue wave aspect**
*by*
*Anatoly Abrashkin and Efim Pelinovsky*

**Reviewer 2:**
First, it suffers a lack of illustration. Indeed, a single figure appears, and intends to show the full geometry of the problem. For instance, from the figure, I cannot understand what is this "average current" (average in time, in 'a' coordinate? In 'b' coordinate?) Neither can I see a weak vorticity. Thus, the definition of vorticity is confusing. Another way to say the same thing is that the Euler to Lagrange coordinate transform is not clear. Is a background vertical flow included? Or do we only consider the vertical flow induced by the waves?

**Authors:**
We are highly appreciated this comment. We shall add the information in the figure. The horizontal current $\psi_{1t_1}$ depends on slow coordinates only. So we name it "average current" as the average in fast variables $a_0, t_0$ (see formula (13)).The weak vorticity is set as an arbitrary function of Lagrange coordinates in some region. For example, the vorticity can differ from zero inside the bounded region. This case corresponds to the interaction of the wave with the localized vortex. **We drew a new figure.** We concentrate on the studying of horizontal current because it is a term of the NLS equation. But in the quadratic approximation there exists the vertical flow too. This one is described by the function $f_2$ depending on the wave amplitude and the horizontal current (see the equation (21)). In the Eulerian coordinates the background flow has two components of the velocity depending on the variables $x, y, t$.

**Reviewer 2:**
Presentation of the results is a little bit confusing.
For instance, it is shown that the absence of vorticity and current, the Akhmediev soliton solution in Lagrange coordinates does correspond to the Akhmediev soliton in Euler coordinates, up to the second order in epsilon. But then, for quadratic and cubic terms, it is claimed the solutions differ. Here could be an interesting result. Could be the authors consider obtaining these solutions and present differencies?

**Authors:**
That is a good idea, but there are two serious problems. Firstly, it is necessary to get a solution up to the third order in Euler coordinates. As we know nobody did that. Secondly, it is necessary to transform our solution to the Eulerian form, i.e. to solve the equations for the functions $\psi_1, \psi_2, \psi_3, f_2, f_3$ and then to express the obtained solutions in Euler variables. This program requires the very unwieldy calculations. So we establish that the realization of the reviewer's idea has to be a subject of a new paper.

**Reviewer 2:**
When considering the Gerstner wave, where does the vorticity profile comes from? Thus, the following sentence is disturbing: "From the physical point of view, this is due to the fact that the average current induced by the vorticity compensates the stokes drift". Is it only true when

integrated? The result associated is very interesting (finding Gerstner waves not affected by modulational instability), but its explanation is not straightforward and should be developed. Still, these waves are a very specific case, and this is not clear from the text.

**Authors:** We added in the next (Sec. 4.2, lines 460-489):
To derive the vorticity of the Gerstner wave Eq. (43) should be substituted in Eq. (6). Then in could be found that in the linear approximation the Gerstner wave is potential ($\Omega_1 = 0$ ), but in the quadratic approximation it possesses vorticity

$$\Omega_{2Gerstner} = -2\omega k^2 |A|^2 e^{2kb}.$$
(44)

For this type of the vorticity distribution the first two terms in the parentheses in Eq. (41) neglect each other. From the physical point of view this is due to the fact that the average current induced by the vorticity compensates the potential drift exactly. The packet of weakly nonlinear Gerstner waves in this approximation is not affected by their non-linearity, and the effect of the modulation instability for the Gerstner wave is absent.

Generally speaking this result is quite obvious. As there are no particle's drift in the Gerstner wave the function $\psi_1$ equals to zero. So the multiplier of the wave's amplitude in Eqs. (38), (39) may be neglected initially without derivation of the vorticity of the Gerster wave.

Let's consider some particular consequences of the obtained result. For the irrotaional ($\Omega_2 = 0$ ) stationary ($A = |A| = const$ ) wave Eq. (40) for the velocity of the drifting flow takes the form

$$\psi_{1t_1} = -\omega kA^2 e^{2kb}.$$
(45)

It coincides with the expression for the Stokes drift in the Lagrangian coordinates (in the Eulerian variables the profile of the Stokes current could be obtained by the substitution of $b$ to $y$ ). Thus, our result may be interpreted as a compensation of the Stokes's drift by the shear flow induced by the Gerstner wave in a square approximation. This conclusion is also fair in the "differential" formulation for vorticities. From Eq. (22) it follows that the vorticity of the Stokes drift equals to the vorticity of the Gerstner wave with the inverse sign.

**Reviewer 2:**
Results of the following part, entitled "Rogue waves", describe the evolution of the coefficients of the NLS equation with the structure of vorticity. In each one of the three cases studied, the eventuality of a breather soliton to exist is analyzed. Maybe, the characteristics of these new Peregrine breather could be described, by comparison with classical one. This analysis would provide an idea of whether or not a vertical flow is amenable to increase the probability of occurrence of rogue waves.

**Authors:** We rewrote the Sec. 4. The differences between potential and vortical wave's solutions are formulated in detail (Sec. 4.3; 4.4, lines 513-592):

[revised manuscript text omitted]

**The authors would like to express our sincere thanks to the reviewer 2 for all the valuable comments and helpful suggestions.**

**Dear Editor!**

**We send the LIST OF CHANGES**
**to the paper "The Lagrange form of the nonlinear Schrödinger equation for low vorticity waves in deep water"** *by Anatoly Abrashkin and Efim Pelinovsky*:

1) **We changed the title of the manuscript;**
2) **We rewrote the introduction to exclude a discussion of the problem of rogue waves;**
3) **We drew the new figure;**
4) **We rewrote the Sec. 4;**
5) **We included a new Sec. 5 which contains the comparative analysis of the Lagrangian and the Eulerian approaches;**
6) **We removed all references with the topic of rogue waves.**

**Sincerely yours,**
**Anatoly Abrashkin and Efim Pelinovsky**

---

## Referee Report (RR1)

The revised version of the manuscript has addressed some of the points I raised in my report. Although I still stand by every word of that report I do not feel that I should force the authors to do something they clearly don't want to do. I see my duty as a reviewer (apart from ensuring that the results are correct) is to help authors and readers in clarifying what has been done, how this relates to the existing knowledge and to provide recommendations to the authors on what else could be done to increase the impact and significance of the findings. Once all that has been made clear, it is up to the authors to decide what they want, since it is their work.

To formulate the comments to the revised manuscript it is helpful first to return to the main points of the discussion. Here are the key point verbatim:

**Reviewer 1:**
Wavetrain modulations upon arbitrary vertically sheared currents were thoroughly studied by Benney and his group. If the Benney asymptotic expansion becomes invalid for the range of small values of vorticity the present work is focusing upon, then it has to be shown and explained what is wrong with the Benney expansion. The same question applies to Jonhson (1976) results. The dependence of the cubic nonlinearity on vorticity in Jonhson (1976) and the works by Benney is not singular. Therefore similar expansion for the small vorticity can be carried out in the Eulerian framework as well using the known results, say, by Jonhson (1976) and/or the works by Benney group as the starting point. **I think what the authors are doing is a re-derivation of the NLS for weak vorticity; the results were known, although implicitly, since nobody looked specifically into this case.**

**Authors:**
**We found a new family of solutions** for the wave train propagation in the deep water. Their novelty is **non-uniform distribution of the vorticity**.

We study flows with the vorticity depending on both Lagrange coordinates. That corresponds to the background current depending on the *x,y, t* variables in the Eulerian approach, not a shear flow *U(y)* as in Johnson or Benney studies. **Our approach differs from other known ones cardinally.**

**The present review:** The following points should be made clear:
(i)     The weak vorticity shear currents are in fact *weak currents* in the appropriately chosen reference frame. In linear setting the effect of weak currents on dispersion relation to leading order in current-to-phase velocity ratio is captured by the Stewart and Joy formula (R. H. Stewart and J. W. Joy., Deep-Sea Res., 21:1039–1049, 1974.). In the NLS context this correction enters the equation additively and yields the term with beta (in the author's notation). Of course, if we apply this literally will get the beta not dependent on the horizontal coordinate. This result is implicit in the NLS derivation by Johnson (1976).
(ii)    However, it is pretty obvious that in the Stewart and Joy formula we can allow the current to depend on slow time or horizontal coordinate. Correspondingly, in the standard (Johnson or Benney) derivations we can take an epsilon square weak current and allow it to depend on epsilon square slow time or space scale.

Thus, the derived NLS with the non-uniform distribution of the vorticity is implicitly contained in the known results. This point should be made clear in the text of the manuscript. On the other hand, this has never been explicitly stated and in this sense is

novel. Perhaps this alternative derivation with emphasis on weak vorticity currents might be of some independent interest.

**Reviewer 1:** The original element of the work is the asymptotic derivation of the NLS in Lagrangian variables. In my view this is complementary to the existing Eulerian works and it remains unclear what new features/aspects this might reveal.

**Authors**: The original aspect of our study is **horizontal non-uniformity of vorticity's distribution** (lines 126-128). As a consequence, in contrast to Benney and his group and Johnson we derived the evolutional equation with variable coefficients.

**The present review**: If you allow the weak shear current to depend either on $T_2$ or $X_2$ the NLS will get the term with beta dependent either on $T_2$ or $X_2$. I reiterate that it has not been explicitly demonstrated.

An important issue of the discussion was about the implications of the derived NLS, about its meaning.

**Reviewer 1:** In contrast to the NLS in Euler variables where we know that the equation describes evolution of the envelope amplitude in the space and how the actual elevation can be expressed as a Stokes-like series in wave amplitude up to cubic order, here the NLS in Lagrangian variables is an object which is much less straightforward to interpret. Obviously, *A* is the envelope amplitude, but what are the independent variables $(a,b)$?

**Authors**: Lagrangian variables are the labels of the fluid particles, nothing more over.

**The present review**: This is not an answer; of course, they are labels. But how do we specify these labels? How can we express the quantites we are interested in in terms of physically well defined variables  In this case a common convention that variables $(a,b)$ are the coordinates at the initial moment does not apply. Hence, there is an issue  which should be recognized. This does not mean that the result itself is meaningless, absolutely not. But one has to formulate a way of expressing the results in well defined variables.

**Reviewer 1:** Their link to the standard Eulerian variables (x,y) is not known. Although, it is straightforward, at least in principle, to provide this link in terms of series in ε, the authors choose not to do this. The effectively use the zero order approximation where the difference between the Eulerian and Lagrangian description vanishes. Then the rationale for using the Lagrangian approach apparently disappears.

**Authors**: **That is not correct.** We derived a new family of solutions due to Lagrange approach. It is much more difficult problem to get them in Eulerian description which has not been solved yet.

**The present review**: I did not see a new family of solutions and I disagree with the author's claim.

**Reviewer 1:** I suspect (this is the most interesting point), that if the authors make transformation to return to the Euler variables, they will get a higher order NLS type equation since the transformation itself is nonlinear (see e.g. F. Nouguier, B. Chapron, C-A, Guérin Second-order Lagrangian description of tri-dimensional gravity wave interactions, JFM 772, 165-196 (2015) and references therein).

**Authors**: That is a special problem. We are ready to discuss it further.

**Reviewer 1:** If the authors do not want to go through this straightforward but quite time consuming pass I suggested above, then they can handle the comparison numerically. The Lagrangian solution yields *X, Y* in terms of *a,b, t*. Hence the surface elevation and position of a parcel on the free surface *Y(a,0,t),* found in terms of series, provide implicit function *Y(a,0,t)* which can be easily plotted for a typical solution, say, a breather. This plot has to be compared to the Eulerian solution with the cubic terms retained.

**Authors**: That is a good programme, but nobody has calculated the Eulerian solution with the cubic terms. All authors are restricted to the derivation of the NLS equation. With what solution do we have to compare our results? Or we must study our problem in Euler variables too? Besides, we are interested in rogue waves in this paper and study the leading order of the solution only. The terms of the second and cubic orders are out of our attention.

**The present review**: To derive NLS or its generalizations in the Eulerian description one has to find the first few terms (up to the cubic) of an asymptotic expansion in epsilon. This has been done by a number of authors. Hence, the surface elevation up to cubic terms in epsilon does exist (at least in implicit form) in the literature. The authors are saying that they are interested in the leading order only. In my view the interesting things will appear in the second and third orders. However, it is up to the authors to decide how far they are prepared to go in this manuscript and what they want to present.

**Conclusion:** If the authors accommodate my comments above, that is they make clear to readers what has been done, I would recommend the publication of this work in the NPG. The English also needs poloshing prior to the publication.

---

## Editor Decision (ED1)

Review of "The Lagrange form of the nonlinear Schrodinger equation for low vorticity waves in deep water: rogue wave aspect" (revised) by Anatoly Abrashkin and Efim Pelinovsky

**Editor Comment**

While the first referee is now satisfied, the second referee has reiterated a major issue which needs to be fully addressed in a further revision. I agree completely with the thrust of the referee's comment, and request that you revise accordingly.

The issue is the relationship between the derived nonlinear Schrodinger equation (NLS) (47) in Lagrangian coordinates and similar extant results in the literature expressed in Eulerian coordinates. Since you have assumed weak vorticity, and the NLS itself is a weakly nonlinear asymptotic expansion, the end result is a simply a phase change, which as the referee has pointed out, can be found in the literature in several places using the more usual Eulerian coordinates. This needs to be fully acknowledged, especially in your section 5, with a full set of appropriate references. I would add that the term "correlation" in the heading of this section would be more appropriately called "equivalnce". Also, as a relatively minor but nonetheless important point, reference to the work of "Benney and his group" needs an explicit reference(s), and finally the English expression can be improved.

1

---

## Author Response (AR3)

**LIST OF CHANGES to the paper "The Lagrange form of the nonlinear Schrödinger equation for low vorticity waves in deep water" by Anatoly Abrashkin and Efim Pelinovsky:**

1) Benny should be Benney throughout.

We corrected the literal error in the family Benney (line 33)

2) The derivation of the NLS by Zakharov was 1967 in the Russian version, and this should be stated.

Lines 33-36: The nonlinear Schrödinger (NLS) equation was first derived by Zakharov in 1967 (English edition, Zakharov, 1968) who used the Hamiltonian formalism for description of wave propagation in deep water; see also Benney and Newell (1967).

3) The reference to Benney et al work on waves on shear flows has been removed. This was not requested. What was asked for was to make this an explicit reference, not just "Benney and his group".

**Lines 51-52:** Wave train modulations upon arbitrary vertically sheared currents were studied by Benney and Maslowe (1975).

**We included a new reference (lines 681-682):**

Benney, D.J. and Maslowe, S.A.: The evolution in space and time of nonlinear waves in parallel shear flows, Stud. Appl. Math., 54, 181–205, 1975.

We would like to thanks Editor and Reviewers for useful comments.

**Sincerely yours, Anatoly Abrashkin and Efim Pelinovsky**